# Hierarchical Clustering Diffusion Model for fMRI Functional Connectivity to Enhance Autism Spectrum Disorder Diagnosis

## Abstract

Functional magnetic resonance imaging (fMRI) data, particularly functional connectivity matrices, are crucial for studying brain disorders like Autism Spectrum Disorder (ASD). However, data scarcity often limits the performance of diagnostic models. We address this challenge by leveraging generative diffusion models for data augmentation. We introduce a novel transformer-based latent diffusion model, the Hierarchical Clustering Connective Diffusion Unit (HC-CDU), designed to synthesize realistic fMRI functional connectivity matrices. Our models effectively generate high-fidelity connectivity patterns, demonstrating an improvement of up to 3.61% in MAE reduction. In classification tasks on the ABIDE-I dataset, HC-CDU with ×1 augmentation demonstrated significant improvement, with AUC enhancing by up to 4.29% over baseline, showcasing enhanced discriminative power.

## 1 Background

Autism spectrum disorder (ASD) is a complex neurodevelopmental condition affecting communication, social interactions and behaviors (1). ASD diagnosis traditionally relies on clinical and behavioral assessments (2). Functional magnetic resonance imaging (fMRI) has emerged as an indispensable tool, offering insights into neural mechanisms underlying ASD via blood oxygen level-dependent (BOLD) signal analysis (3).

Machine learning has enabled numerous automated diagnostic frameworks based on functional connectivity analysis (6; 7). However, optimal performance requires large datasets. Collecting high-quality fMRI data is expensive, time-consuming and requires specialized equipment, leading to data scarcity that hampers ML model performance and generalizability.

fMRI data augmentation has explored traditional approaches such as adding Gaussian noise (8) or sliding windows on ROI signals (9). Sophisticated methods using Variational AutoEncoders (VAE) (10) and Generative Adversarial Networks (GAN) (11; 12) synthesize data but suffer from "posterior collapse" or "mode collapse" problems. Diffusion models (13; 14; 15) have emerged as powerful alternatives, demonstrating remarkable ability to generate high-quality medical imaging data.

## 2 Methods

The proposed HC-CDU is a transformer-based latent diffusion model enhanced by hierarchical clustering, as illustrated in Figure 1. For comparison, we also evaluate a non-hierarchical variant (CDU), which omits the hierarchical conditioning module. HC-CDU generates diverse and realistic fMRI functional connectivity matrices through three main components.

## 2.1 Latent Connectivity Autoencoder

We adopt a Variational Quantized Autoencoder (VQ-VAE) to encode and discretize functional connectivity matrices into a structured latent space (16). The architecture consists of an encoder that maps a functional connectivity matrix $F \in \mathbb{R}^{V \times V}$ (where $V$ is the number of ROIs) to a latent representation $\zeta_{enc}$. Implemented as a CNN with convolutional layers and residual blocks, it projects $F$ into a feature space of dimension $D_{lat} \times H_{lat} \times W_{lat}$. Each row of $F$ represents one ROI's correlation with all others.

A vector quantization module discretizes the latent features, mitigating posterior collapse issues. A codebook $\mathcal{K} \in \mathbb{R}^{C \times D_{lat}}$ with $C$ learnable vectors maps each latent feature $z_{enc,i} \in \zeta_{enc}$ to its closest codebook entry:

$$z_{quant,i} = \kappa_j, \quad where j = \arg\min_k \|z_{enc,i} - \kappa_k\|^2 \tag{1}$$

A decoder reconstructs the connectivity matrix $F' \in \mathbb{R}^{V \times V}$ from $\zeta_{quant}$ using deconvolutional layers. The latent autoencoder is trained by minimizing:

$$L_{VAE} = L_{recons} + L_{VQ} + L_{comm} \tag{2}$$

## 2.2 Conditional Diffusion Transformer with Hierarchical Conditioning

After training the latent connectivity autoencoder, learned latent representations $\zeta_{enc}$ serve as inputs to a conditional diffusion transformer, trained as a noise prediction network $\eta_\psi$. In HC-CDU, this model is conditioned on the subject's diagnostic state $c$, diffusion timestep $k$, and multi-scale hierarchical cluster embeddings derived via an integrated Hierarchical Clustering module, which performs two-level clustering to extract representations capturing functional brain organization at multiple scales.

The forward diffusion process progressively adds Gaussian noise to an initial latent representation $\zeta_0$ over $K$ time steps:

$$q(\zeta_k \mid \zeta_0) = \mathcal{N}(\zeta_k; \sqrt{\alpha_k}\zeta_0, (1 - \alpha_k)I) \tag{3}$$

where $\alpha_k$ is a scaling factor dependent on time step $k$.

The inverse diffusion process systematically denoises $\zeta_k$ to recover $\zeta_0$ through a transformer-based noise prediction network. The architecture utilizes transformer blocks that predict noise present in $\zeta_k$. To incorporate conditioning information (timestep $k$, class label), we employ adaptive layer normalization (adaLN):

$$adaLN(\zeta_k, k, C_{cond}) = \gamma(k, C_{cond}) \cdot \frac{\zeta_k - mean(\zeta_k)}{std(\zeta_k)} + \beta(k, C_{cond}) \tag{4}$$

The noise prediction network is trained by minimizing the L1 loss between predicted and true noise:

$$L_{diffusion} = \mathbb{E}_{\zeta_0, \eta, k}[\|\eta_k - \eta_\psi(\zeta_k, k, C_{cond})\|^2] \tag{5}$$

New functional connectivity matrices are generated by integrating the trained VQ-VAE and conditional diffusion transformer, modulated by hierarchical cluster embeddings. Starting from initial noise and target class label, the model iteratively denoises via reverse diffusion process, with hierarchical conditioning applied at each step.

# 3 Experiments and Results

**Dataset:** We developed and evaluated our methodology using resting-state fMRI data from the ABIDE-I dataset (17), with 505 ASD and 530 controls. The brain was segmented into 200 ROIs using the CC200 atlas (18). We adopted a stratified 5-fold cross-validation strategy with approximately 60% training, 20% validation, and 20% testing data.

**Implementation:** VQ-VAE trained 800 epochs (batch size 64, learning rate 3e-4, commitment cost 0.25). Architecture: 128 internal channels, 2 residual blocks, embedding dim 16, codebook size 768. Diffusion models trained 600 epochs (batch size 4, learning rate 3e-4, 100 timesteps, L1 loss).

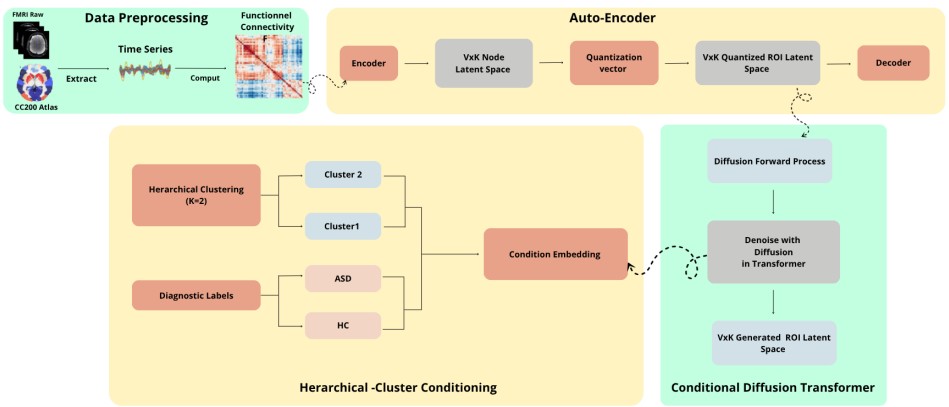

Figure 1: The comprehensive HC-CDU pipeline includes: (1) Data preprocessing to extract ROI signals and compute functional connectivity matrices; (2) a Latent Connectivity Autoencoder (VQ-VAE) that encodes inputs into structured latent space; and (3) a Conditional Diffusion Transformer that generates novel latent representations, guided by hierarchical clustering to capture multi-scale brain network structures.

Table 1: 5-Fold Cross-Validation Performance

| Fold | No Augmentation | | | | HC-CDU x1 | | | | HC-CDU x4 | | | | CDU x1 | | | | CDU x4 | | | |
|---|---|---|---|---|---|---|---|---|---|---|---|---|---|---|---|---|---|---|---|---|
| | Acc | AUC | Sen | Spe | Acc | AUC | Sen | Spe | Acc | AUC | Sen | Spe | Acc | AUC | Sen | Spe | Acc | AUC | Sen | Spe |
| 1 | 69.57 | 74.19 | 73.58 | 65.35 | **71.50** | **78.48** | **76.42** | **66.34** | 67.15 | **75.39** | 79.25 | 54.46 | 69.08 | **76.66** | 72.64 | 65.35 | 64.25 | 72.86 | **77.36** | 50.50 |
| 2 | 68.12 | 76.78 | 70.75 | 65.35 | **70.05** | **79.28** | **71.70** | **68.32** | **69.08** | 75.55 | **73.58** | 64.36 | **69.08** | **76.81** | 67.92 | **70.30** | 65.22 | 73.21 | **77.36** | 52.48 |
| 3 | 67.63 | 72.70 | 64.15 | 71.29 | 67.63 | **73.68** | **66.98** | 68.32 | 59.90 | 66.91 | **79.25** | 39.60 | 66.18 | **73.70** | **66.98** | 65.35 | 59.42 | 66.97 | 40.59 | **77.36** |
| 4 | 70.53 | 76.26 | 71.70 | 69.31 | 61.35 | 69.48 | 64.15 | 58.42 | 62.32 | 66.58 | **72.64** | 51.49 | 62.32 | 69.88 | 67.92 | 56.44 | 60.39 | 66.77 | **79.25** | 40.59 |
| 5 | 66.18 | 74.39 | 67.92 | 64.36 | **67.63** | **75.68** | **70.75** | 64.36 | **66.67** | **75.74** | 69.81 | 63.37 | **66.67** | 74.38 | **70.75** | 62.38 | 66.18 | 74.25 | **81.13** | 50.50 |

Table 2: Mean Absolute Error Assessment for Generated Functional Connectivity Matrices

| Scenario | MAE (Mean $\pm$ Std) |
|---|---|
| Real Data Baseline | 0.194 $\pm$ 0.001 |
| HC-CDU x1 | **0.187 $\pm$ 0.003** |
| HC-CDU x4 | **0.187 $\pm$ 0.003** |
| CDU x1 | **0.177 $\pm$ 0.001** |
| CDU x4 | **0.178 $\pm$ 0.001** |

Transformer: hidden size 128, depth 14, 8 heads. HC-CDU used 20 and 8 clusters with temperature 0.5. SVM classifier with RBF kernel for evaluation.

HC-CDU ×1 (Table 1) achieved balanced improvements. Fold 1: 78.48% AUC (+4.29% over baseline). Average: 67.63% accuracy, 75.32% AUC. CDU ×1: 74.29% AUC, 66.67% accuracy. Higher augmentation (×4) reduced performance but increased sensitivity. All models (Table 2) achieved lower MAE than baseline, confirming high-fidelity synthetic data generation.

# 4 Conclusion

We introduced HC-CDU, a hierarchical clustering-enhanced diffusion model for fMRI connectivity augmentation in ASD diagnosis. Moderate synthetic data augmentation significantly improves diagnostic performance while maintaining data fidelity. Hierarchical clustering provides benefits over non-hierarchical approaches, establishing a framework for addressing data scarcity in psychiatric neuroimaging research.

# 5 Broader Impact

Potential negative impacts include privacy concerns from synthetic data generation, bias amplification, and over-dependence on automated tools. We emphasize careful validation and human oversight.

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
