# OpenReview forum: "Hierarchical Clustering Diffusion Model for fMRI Functional Connectivity to Enhance Autism Spectrum Disorder Diagnosis"
_EurIPS.cc/2025/Workshop/MedEurIPS — EurIPS 2025 Workshop MedEurIPS Submission_

### Official Review · Reviewer_RhY8 · 2025-10-24
**Good workshop paper, strengthen for publication**

**Rating:** 6
**Confidence:** 4

**Review:**

This paper presents a data augmentation strategy for functional connectivity matrices using a latent diffusion model with hierarchical conditioning to improve autism spectrum disorder diagnosis. The idea is interesting and fits well within the scope of the workshop. However, there are several issues that should be addressed to strengthen this work:

- The authors do not sufficiently explain how hierarchical clustering actually works in their framework.
- The “Experiments and Results” section lacks analysis: there is no discussion of what the evaluation metrics represent or what insights can be drawn from the results. I also recommend presenting averaged results across folds (rather than showing individual folds) for clarity.
- Minor issue: Equation (5) shows an L2 loss, not L1 as stated.

Finally, I strongly encourage the authors to include an additional experiment measuring codebook utilization in their VQ-VAE model, as codebook collapse can be a significant issue for this architecture.

---

### Official Review · Reviewer_PYB6 · 2025-10-31

**Rating:** 5
**Confidence:** 4

**Review:**

This work aims to address the challenge of limited fMRI data for training machine learning models for autism spectrum disorder (ASD) diagnosis. Overall, the paper is well motivated: collecting high-quality fMRI data is expensive and time-consuming, while performing data augmentation on connectivity matrices is not straightforward. The authors employ a generative data-augmentation approach using a class-conditional latent diffusion model with a transformer-based denoiser, and use hierarchical cluster embeddings as an additional conditioning signal to capture multi-scale brain network structure.

The authors report results on the ABIDE-I dataset, which includes 505 ASD subjects and 530 controls, using a 5-fold cross-validation evaluation strategy. The presented approach appears to outperform a conditional latent diffusion model conditioned only on diagnostic state (class label). However, the description of the experimental setup is very limited, making it difficult to fully understand the effect of the additional condition and to draw clear conclusions. The presented approach tailored to fMRI data is interesting, though the technical novelty appears relatively limited, I believe. Clarifying the data-generation and evaluation protocol in greater depth would further improve the clarity and impact of the work.

Have the authors performed any evaluation to assess the realism of the generated connectivity matrices? Have the authors considered diffusion-driven counterfactual generation for fMRI [A] as an alternative or complementary augmentation strategy?

[A] Bedel, Hasan A., and Tolga Çukur. "Dreamr: Diffusion-driven counterfactual explanation for functional MRI." IEEE Transactions on Medical Imaging (2024).

---

### Decision · Program_Chairs · 2025-11-03

**Decision:**

Reject

**Comment:**

The paper proposes a generative data augmentation strategy for fMRI, using a hierarchically conditioned Latent Diffusion Model for ASD diagnosis. However, the work does not meet the acceptance bar due to critical methodological shortcomings. The core technical novelty is limited, and the authors fail to provide a sufficient technical explanation of the hierarchical conditioning mechanism. Furthermore, the evaluation lacks rigor, as essential validation experiments, including crucial component ablations and comparisons against recent state-of-the-art baselines, are missing.